# OpenReview forum: "Derail Yourself: Multi-turn LLM Jailbreak Attack through self-discovered clues"
_ICLR.cc/2025/Conference — ICLR 2025 Conference Withdrawn Submission_

### Official Review · Reviewer_2ps2 · 2024-10-30

**Soundness:** 2
**Presentation:** 3
**Contribution:** 3
**Rating:** 5
**Confidence:** 5

**Summary:**

This paper introduces ActorAttack, a method that utilizes actor-network theory to generate diverse attack paths while concealing harmful intents. Unlike existing methods, ActorAttack automates the discovery of attack clues using LLM knowledge, enhancing both the effectiveness and diversity of attacks. The research also presents the SafeMTData dataset, which improves LLM robustness through safety tuning, revealing a trade-off between utility and safety. This work emphasizes the need for addressing safety risks in multi-turn conversations with LLMs.

**Strengths:**

- The proposed method demonstrates a better attack success rate.
- The manuscript discusses its helpfulness, and further fine-tuning indicates that the proposed method is effective in enhancing model security.

**Weaknesses:**

- Weak Technical Contribution. From my perspective, the primary contribution of the proposed attack is the identification of attack clues derived from LLMs. However, there is a lack of quantitative analysis regarding the superiority of these clues compared to existing works, such as CFA [1] and CoA [2]. Why are these attack clues more effective? Is it due to their ability to introduce a greater range of possibilities into the attack chain? It is important to note that existing works also leverage related problems to enhance their attack chains, allowing for the incorporation of additional possibilities as they evolve.
- Datasets: The number of samples in the dataset is unclear. I only see that the ablation study used 50 samples. This raises concerns about the generalizability of the proposed method.
- Baseline: The proposed method involves multi-turn jailbreaking, so the comparison should focus on multi-turn attacks as well. However, the baseline presented only includes one multi-tuen jailbreaking method, Crescendo, while other methods like CFA and CoA are not compared. In addition, the chosen single-turn jailbreak methods are not state-of-the-art, as more effective attacks (such as AutoDAN, ReNeLLM, and MultiLingual) are not compared. For details, see: http://easyjailbreak.org/leader-board.
- Incomplete Evaluation. I am particularly interested in the comparison between the proposed method and multi-turn jailbreaking. However, the authors have limited the evaluation of victim models to only GPT-4o and Claude-3.5. Why not conduct a comprehensive evaluation across all five victim models listed in Table 1? This suggests a potential selective presentation of results by the authors.
- Efficiency Evaluation. The proposed method relies on LLMs to generate multiple attack clues, which may be time-consuming. I am concerned that it may be less efficient than existing methods, so the authors should discuss the time costs associated with the attacks.

[1] Multi-Turn Context Jailbreak Attack on Large Language Models From First Principles.
[2] Chain of Attack: a Semantic-Driven Contextual Multi-Turn attacker for LLM.

**Questions:**

See weaknesses. While my current evaluation is negative, I would reconsider the score if the author addresses my concerns, especially by clearly explaining the importance of the attack clues.

---

> ### Author Response · Authors · 2024-11-24
> **Author Rebuttal (Part one)**
>
> Thank you for your thoughtful review and for recognizing the strengths of our work! We are pleased to hear that you found the effectiveness of our proposed method and appreciated its utility in enhancing model safety. We have addressed your concerns below and are happy to continue the discussion to further clarify our contributions.
>
> Q1: *Our contributions over multi-turn baselines:*
>
> A1: Thank you for your thoughtful observation. However, we respectfully disagree with the concern about the novelty and technical contribution of our work. **Grounded in sociological theory, our approach introduces a novel and systematic perspective to identify attack clues, enabling more diverse, effective, and efficient attacks** compared to existing multi-turn baselines.
>
> Key Contributions and Novelty
>
> We address two critical challenges for successful multi-turn attacks, which differentiate our work from prior approaches:
> 1. **Identifying Semantically Relevant and Diverse Attack Clues**
>   - Existing approaches either rely on limited strategies like role-playing and scenario assumptions (CoA [1] and CFA [2]) or fixed communication templates plus human-crafted seed instances (Crescendo [3]) to generate attacks. Limited attack strategies and potential biases towards seed instances may lead to the diversity issue, ultimately limiting their effectiveness (Table 1, and Figures 7 and 8).
>   - In contrast, our approach is systematically supported by scientific theory. Inspired by Latour’s sociological network theory, we explicitly model the potential attack clues as a network:
>     - **Nodes** represent attack clues that are semantically relevant to the harmful target.
>     - **Edges** form stepwise attack paths connecting clues to the harmful target.
>     - Our network has two distinct properties, which ensure the effectiveness of our attacks:
>       - **Higher diversity**: we identify six distinct types of nodes, including both human and non-human entities, covering a broader spectrum of attack clues.
>       - **Inherent semantic relevance**: different from existing works that infer attack clues from the seed instances, we generate a unique network for each harmful target, ensuring the derived clues are semantically relevant to the given target (Figure 2 and Figure 3(a)).
>     - Our quantitative results (Table 1 and Table 4) demonstrate the enhanced diversity and effectiveness of our approach.
> 2. **Efficient Convergence Toward the Harmful Target**
>   - Existing works like Crescendo reactively adjusts attack paths during the attack process based on the target model's responses. By contrast, we pre-compute the attack paths (i.e., each edge of our network serves the attack path) and the inherent semantic relevance of our attack clues ensures the convergence of our attack paths.
>   - Experimental results show that **our multi-turn attacks achieve high effectiveness without relying on feedback from the target model** (Table 1), validating the efficiency of our method. Another evidence is that under the same attack budget, our attacks consistently achieve greater effectiveness than Crescendo  (Figure 4). We also conducted a comparison between CoA and our method (see the table in response to Q5). These results demonstrate the efficiency of our method.
>
> In summary, our work introduces a novel perspective and addresses the limitations of previous methods, achieving state-of-the-art performance in multi-turn attacks. We hope this explanation clarifies our contributions and the distinction from multi-turn baselines.

---

> ### Author Response · Authors · 2024-11-24
> **Author Rebuttal (Part two)**
>
> Q2: ***Comparison with more multi-turn baselines:***
>
> A2: Following the reviewer’s suggestion, we evaluated CoA on Harmbench. However, we did not include CFA in our comparison because it is not open-sourced, and key implementation details (such as the system prompt) are unavailable. The results, as shown in the table, further validate the effectiveness of our ActorAttack, demonstrating superior performance over CoA against both open-source and closed-source models.
>
> |              | Llama-3-8b-Instruct | GLM4-9b-instruct | Qwen2.5-7b-instruct | GPT-4o   | Claude-3.5-sonnet |
> |:--------------:|:---------------------:|:------------------:|:---------------------:|:----------:|:-------------------:|
> | CoA          | 25.5                | 26.0             | 25.0                | 17.5     | 3.4               |
> | ActorAttack  | **79.0**            | **73.0**         | **83.5**            | **84.5** | **66.5**          |
>
> Q3: ***Incomplete evaluation in ablation study***
>
> A3: We appreciate the reviewer’s concern regarding the target models used in the ablation study. Here, we clarify the rationale behind our choices and provide additional evaluations to address the concern:
>   1. Rationale for Model Selection: We selected GPT-4o and Claude-3.5 as representative victim models because these LLMs are generally safer than alternatives like Vicuna, Qwen, or ChatGLM, as evidenced by their superior performance on safety benchmarks such as Harm-bench and Sorry-bench.
>   2. Additional Evaluations: To further address your concern, we conducted a more comprehensive evaluation to demonstrate the generality of our ActorAttack approach. Specifically, we included three additional models: Llama-3-8B-instruct, Qwen-2.5-7B-instruct, and GLM-4-9b-chat. In these experiments, we explored how varying the number of our attack clues affects effectiveness. The results, consistent with those in our paper, show that the proportion of successful attacks increases as the number of actors (attack clues) grows. It indicates that our ActorAttack can find more optimal attacks from our diverse attack paths. We will include these additional findings in the camera-ready version of our paper.
>
> | Number of attack clues | Llama-3-8b-instruct | Qwen2.5-7b-instruct  | GLM4-9b-instruct  |
> |:----------------------:|:-------------------:|:--------------------:|:-----------------:|
> |            1           |         62.0        |         65.0         |        54.0       |
> |            2           |         74.0        |         79.0         |        69.0       |
> |            3           |         79.0        |         83.5         |        73.0       |
> |            4           |         84.5        |         86.0         |        74.0       |
> |            5           |         85.0        |         87.5         |        76.5       |
>
> Q4: ***Comparison with more single-turn baselines.***
>
> A4: Thank you for the suggestion! We have added the results of ReNeLLM and MultiLingual to the table below. Our attack demonstrates stronger effectiveness compared to these single-attack baselines. In particular, these state-of-the-art single-attack baselines, including PAIR, PAP, and CipherChat used in our paper, perform poorly on Claude-3.5-sonnet, while our attack achieves a much higher success rate. It highlights the severity and pervasive nature of the safety vulnerabilities we identified. Due to the high computational cost of evaluating AutoDAN, its results are not yet complete. We will update this as soon as possible.
>
> |              | Llama-3-8b-instruct | Qwen2.5-7b-instruct | GLM4-9b-instruct |  GPT-4o  | Claude-3.5-sonnet | Average  |
> |:------------:|:-------------------:|:-------------------:|:----------------:|:--------:|:-----------------:|:--------:|
> |   ReNeLLM    |         76.0        |         69.0        |       68.5       |   68.0   |        24.5       |   61.2   |
> | MultiLingual |          0          |        100.0        |       100.0      |     0    |         0         |   40.0   |
> | ActorAttack  |       **79.0**      |       **83.5**      |     **73.0**     | **84.5** |      **66.5**     | **77.3** |

---

> > ### Author Response · Authors · 2024-11-24
> > **Author Rebuttal (Part three)**
> >
> > Q5: ***Efficiency evaluation.***
> >
> > A5: Thank you for your suggestion! We respectfully address your concerns regarding the efficiency comparison. We conducted several experiments to highlight the efficiency advantages of our method in our paper:
> > 1. High Success Rates Without Feedback: Our multi-turn attacks achieve high success rates without requiring feedback from the target model (as shown in Table 1).
> > 2. Better Performance Within Attack Budget: Under the same attack budget, our method consistently achieves a significantly higher attack success rate compared to Crescendo (as shown in Figure 4).
> >
> > To fully address your concern, we included comparisons with CoA. However, due to differences in backends (e.g., vLLM or Torch) and parallelization mechanisms used by these methods, it is challenging to directly compare time costs fairly. Instead, we propose using the average number of interactions with the target model per query as an efficiency metric. In this context, each turn of a query, including random trials, is counted as one interaction. The results in the table below show that our method generally requires fewer interactions to succeed compared to CoA, further demonstrating the efficiency advantages of our approach.
> >
> > |             | Llama-3-8b-instruct | Qwen2.5-7b-instruct | ChatGLM4-9b-instruct | GPT-4o | Claude-3.5-sonnet | Average |
> > |:-----------:|:-------------------:|:-------------------:|:--------------------:|:------:|:-----------------:|:-------:|
> > |     CoA     |         14.6        |         13.9        |         14.2         |  14.6  |        43.3       |   20.1  |
> > | ActorAttack |         8.3         |         7.9         |          9.4         |   8.1  |        10.9       |   11.2  |

---

> > ### Comment · Reviewer_2ps2 · 2024-11-25
> > **Official Comment by Reviewer 2ps2**
> >
> > Thank you for your efforts in conducting numerous additional experiments.
> >
> > Regarding  A3, I feel that you may not have fully understood my question. Why are the more comparable multi-turn jailbreaking baselines treated as ablation studies? This setting seems particularly inappropriate, and it should instead be included in Table 1 as a standard comparison. Moreover, the introduction of additional victim models such as GLM and Qwen seems unnecessary. The five victim models already presented in Table 1 are sufficient for evaluation. Ensuring consistency in comparisons across these models would provide clearer and more meaningful results.
> >
> > Regarding A4, the results confirm my earlier point that ReNeLLM and MultiLingual are indeed stronger baselines. However, it is crucial to ensure that the victim models are consistent with those presented in Table 1 of the paper. Additionally, I noticed that MultiLingual exhibits cases with an ASR of 0. This is surprising, as the original paper reports notable performance on both ChatGPT and GPT-4. Please verify your reproduction of these experiments to address this discrepancy.

---

> > > ### Author Response · Authors · 2024-11-25
> > > **Responses to reviewer's feedback**
> > >
> > > First of all, we would like to thank you for your recognition of our experimental work and for your timely feedback. We are grateful for the opportunity to address your two important questions:
> > >
> > > Q1: ***Ensuring consistency of comparisons with baselines.***
> > >
> > > A1: You are right that consistency in comparing our method with both single-turn and multi-turn baselines is crucial. In response to your suggestion, we have conducted a thorough evaluation of the recommended baselines using the same victim model as in Table 1. The updated results, presented in the table below, demonstrate that our ActorAttack achieves the highest attack effectiveness, whether at the level of individual models or on average. This highlights the severe and pervasive safety vulnerabilities in the current LLMs we identified.
> > >
> > > We will incorporate these updated results into Table 1 in camera-ready. Additionally, experiments with AutoDAN on the Llama-3-70B-instruct model are still ongoing, and we will update those results as soon as they are available.
> > >
> > > |                    |               | Llama-3-8B-instruct | Llama-3-70B-instruct | GPT-3.5-turbo-1106 |  GPT-4o  | Claude-3.5-sonnet | Average  |
> > > |:------------------:|:-------------:|:-------------------:|:--------------------:|:------------------:|:--------:|:-----------------:|:--------:|
> > > |                    | Multilingual  |         0.0         |          0.0         |        64.0        |    0.0   |        0.0        |   12.8   |
> > > | Single-turn attack |    ReNeLLM    |        76.0         |         69.5         |        55.0        |   68.0   |        24.5       |   58.6   |
> > > |                    |    AutoDAN    |         37.5        |           -          |          -         |     -    |         -         |     -    |
> > > |                    |      CoA      |         25.5        |         18.8         |        15.5        |   17.5   |        3.4        |   16.14  |
> > > |  Multi-turn attack |   Crescendo   |         60.0        |         62.0         |        38.0        |   46.0   |        50.0       |   51.2   |
> > > |                    |  ActorAttack  |       **79.0**      |       **85.5**       |      **78.5**      | **84.5** |      **66.5**     | **78.8** |
> > >
> > > Q2: ***Discrepancy between our evaluation results of MultiLingual and its original results.***
> > >
> > > A2: Thank you for your observation! We believe the discrepancy arises from the fact that, although we used the same or similar GPT model names, we evaluated different versions compared to [1]. Specifically, we found that newer versions of the GPT model appear to be more robust against the MultiLingual attack.
> > >
> > > For example, [1] used the GPT-3.5-turbo-0613 and GPT-4-0613 models in their evaluation, where MultiLingual achieved higher attack success rates. In contrast, in our paper, we used the more advanced versions, GPT-3.5-turbo-1106 and GPT-4o (a multimodal version), as the target models, and we observed a significant reduction in MultiLingual's effectiveness.
> > > We speculate that this may be because model developers have recognized advanced jailbreak attacks, such as MultiLingual, and have progressively implemented adaptive defenses in their newer models.
> > >
> > > Thank you once again for your valuable suggestions! We look forward to continuing our discussion!
> > >
> > > |              | GPT-3.5-turbo-0613 | GPT-3.5-turbo-1106 | GPT-4-0613 | GPT-4o |
> > > |:------------:|:------------------:|:------------------:|:----------:|:------:|
> > > | Multilingual |        100%        |         64%        |    30.33   |   0.0  |
> > >
> > >
> > > References:
> > > 1. Yue Deng, Wenxuan Zhang, Sinno Jialin Pan, and Lidong Bing. Multilingual jailbreak challenges in large language models. arXiv preprint arXiv:2310.06474.

---

> > > > ### Comment · Reviewer_2ps2 · 2024-11-27
> > > > **Official Comment by Reviewer 2ps2**
> > > >
> > > > Thank you for providing the additional experimental results. They address my concerns regarding the experiments. Furthermore, I maintain my opinion on the technical contributions (Reviewer 8h4V also shared similar concern). The authors' response mentions two properties: Higher Diversity and Inherent Semantic Relevance.
> > > >
> > > > However, my concerns are as follows (using CoA as an example):
> > > >
> > > > - First, I acknowledge that the attack clues in ActorAttack contribute to the diversity of the attacks. However, CoA also initializes the attack chain using an LLM, and if they initialize more attack chains, they would exhibit similar diverse attack chains.
> > > >
> > > > - Inherent Semantic Relevance is not unique to ActorAttack. CoA, while generating attack chains with the LLM, also enforces semantic relevance.
> > > >
> > > > Although ActorAttack has demonstrated its superiority in the experiments, I believe the technical improvements are incremental. Given that ICLR is a top-tier conference, I still find it slightly below the acceptance threshold. Therefore, given the authors' efforts during the response period in addressing the experiments, as well as the inherent technical limitations, I have only raised my score to 5.

---

> > > > > ### Author Response · Authors · 2024-11-28
> > > > > **Thank you for raising your score!**
> > > > >
> > > > > Thank you for your feedback and for raising your score! It seems that we may not have clearly conveyed the novelty of our approach, which may have led to some misunderstandings. We’d like to take this opportunity to clarify our contributions.
> > > > >
> > > > > 1. ***Addressing Diversity in Attack Strategies***
> > > > >
> > > > > First, CoA (and other baseline methods) have limitations in terms of diversity, which our method overcomes. These approaches either rely on fixed strategies like role-playing and scenario assumptions (CoA [1] and CFA [2]) or fixed communication templates plus human-crafted seed instances (Crescendo [3]) to generate attacks.
> > > > >
> > > > > Repeating fixed strategies in each trial and potential biases towards seed instances can lead to a diversity issue, ultimately limiting their effectiveness (see Figures 7 and 8).
> > > > >
> > > > > For example, in the implementation, CoA performs up to 20 random trials per harmful query, which is far more than our ActorAttack's 3 attempts. Despite this, CoA's performance on models like GPT-4o and Claude-3.5-sonnet is significantly lower than ours (e.g., GPT-4o: 84.5% vs. 17.5%, Claude-3.5-sonnet: 66.5% vs. 3.4%), indicating that their limited attack strategies can be easily defended. This gap in performance highlights the novelty of our approach: by introducing a systematic way to identify diverse attack clues grounded in sociological theory, we are able to produce more diverse and effective attacks.
> > > > >
> > > > > 2. ***Our inherent semantic relevance brings an efficiency advantage***
> > > > >
> > > > > You are correct that CoA also ensures semantic relevance during attack generation. However, our approach differs significantly in how we achieve and benefit from this semantic relevance. Existing methods require reactively adjusting attack paths based on the target model's responses, during the attack process.
> > > > >
> > > > > In contrast, our method pre-computes the attack paths, where each edge of our network serves an attack path. This allows us to guarantee semantic relevance in our attack paths **prior to** engaging with the victim model. Our design brings the efficiency advantage and is demonstrated in our experimental results, where our multi-turn attacks achieve high effectiveness without requiring interactive model feedback, as shown in Table 1. Moreover, we further highlight the efficiency advantage in our time-cost analysis (see Table in A5).
> > > > >
> > > > > 3. ***Contribution Beyond Attack Strategies: The Safety Alignment Dataset***
> > > > >
> > > > > In addition to the technical novelty of our attack method, we believe that our contribution extends to the broader safety alignment community. Specifically, we have developed the first multi-turn safety alignment dataset, which was created based on our attack methodology. This dataset, when used to fine-tune LLMs, has shown to greatly improve their robustness against multi-turn attacks, providing valuable insights into how models can be better aligned to handle complex multi-turn attack scenarios.
> > > > >
> > > > > We believe this contribution is highly relevant to the safety alignment community and represents a meaningful step forward in the field.
> > > > >
> > > > > 4. ***Conclusion***
> > > > >
> > > > > In summary, we hope that the additional details we've provided help clarify the novelty and significance of our work. Our approach is not just an incremental improvement, but a step toward addressing the inherent limitations of current attack strategies in terms of both diversity and efficiency. We invite you to revisit our technical contributions with these points in mind and would be happy to continue discussing any remaining concerns you may have.
> > > > >
> > > > > Thank you again for your thoughtful review and your willingness to engage in further discussion. We look forward to your feedback.

---

> > > > > > ### Comment · Reviewer_2ps2 · 2024-11-29
> > > > > > **Official Comment by Reviewer 2ps2**
> > > > > >
> > > > > > Thank you for your response. After carefully considering your rebuttal and the feedback from the other reviewers, I believe the current score is appropriate.

---

### Official Review · Reviewer_uLeg · 2024-10-30

**Soundness:** 3
**Presentation:** 3
**Contribution:** 3
**Rating:** 6
**Confidence:** 4

**Summary:**

Summary:
This paper proposes a multi-turn LLM attack method and addresses two challenges in this method: (1) conceal of harmful intent (2) generating multiple attack paths for the same harmful target. This work adapts the actor-network theory to generate attack clues, which in turn are used to generate concrete prompts which when input to LLMs lead to jailbreak. These carefully generated prompts are designed in a manner that lead to the LLM answering the original harmful query. The work develops a dataset with multi-turn adversarial prompts and safety-alignment data, which can be used for fine-tuning.

**Strengths:**

1. Novel adaptation of actor network theory in multi-turn jailbreak prompt sequence generation.
2. Attack method is suitable for both black-box and white-box LLMs.

**Weaknesses:**

1. Method section 3 could have been written in a better way. Actor network adaptation to the generation of attack clues could be explained in a better way.
2. Uses 4 LLMs as per the algorithm pseudocode. However, clear description of all the 4 models is missing.
3. In Table 1, ASR comparison is done against single turn attacks only. Need to compare with recent related work on Multi-turn attacks like Crescendo, etc.
4. Notations used in algorithm have not been explained in text. This needs to be done for better understanding of the method.
5. Missing experiments for comparison of state-of-the-art defense mechanisms. Experiments section needs to be refined.

**Questions:**

1. What is the time required to generate a jailbreak attack sequence using ActorAttack method? Since, it is using 4 different models, how can it be used in practical use cases?
2. How to interpret the results shown in figures 4(a) and 4(b)? Please add additional details in caption.

---

> ### Author Response · Authors · 2024-11-26
> **Author Rebuttal (Part one)**
>
> Thank you for your thoughtful and encouraging reviews! We are pleased to hear that you recognized the novelty of our adaptation of actor-network theory in multi-turn jailbreaks, and our method’s applicability to both black-box and white-box LLMs. We have addressed your concerns below and look forward to further discussion to refine and clarify our contributions. Additionally, we have uploaded the revised version of our paper, incorporating the changes based on your feedback.
>
> Q1: ***Improving the writing of section 3***
>
> A1: Thank you for pointing out the writing issues in Section 3! To improve the clarity of the process by which we construct the actor network to discover attack clues, we have restructured Section 3.1 (line 192) into three distinct parts:
> - Theoretical Grounding in Our Design
> - Network Definition
> - Network Adaptation to New Harmful Targets
>
> Specifically, each node in the network represents an attack clue. For each harmful target, we generate a unique network, ensuring that the derived clues are semantically relevant to the given target. Our updated descriptions should make the process clearer to readers.
>
> Q2 & Q4: ***Descriptions of notations used in algorithm pseudocode***
>
> A2: Thank you for your valuable feedback! To improve clarity, we have added a detailed description of the notations used in the algorithm pseudocode in the revised manuscript (line 160). Specifically, we clarify the roles of the four LLMs and provide explanations for symbols that were not previously introduced in the main text.
> - The victim model $V_{\theta}$ represents the model being attacked.
> - The attacker model $A_{\theta}$ generates multi-turn attacks.
> - The judge model  $J_{\theta}$ determines the success of the attack.
> - The monitor model $M_{\theta}$ is responsible for our dynamic modification, as illustrated in Figure 3(c).
>
> Additionally, except for the victim model, we employ the same LLM to implement the other three models. This clarification should make the relationships and roles of these models clearer to the readers.
>
> Q3: ***Consistency of comparison with single-turn and multi-turn attack baselines***
>
> A3: Thank you for your helpful suggestion. In Figure 4, we compared the attack success rates of our method and Crescendo for GPT-4o and Claude-3.5-sonnet. However, we agree that comparing single-turn and multi-turn attack baselines within the same setup would make the results clearer and more meaningful.
>
> In response, we have conducted a more thorough evaluation. This includes a new multi-turn baseline, CoA [1], which was suggested by other reviewers. The updated results, shown in the table below, demonstrate that our method, ActorAttack, achieves the highest attack effectiveness, both for individual models and on average. This highlights the significant safety vulnerabilities we have identified in current LLMs. We will update Table 1 in the camera-ready version to include these new results.
>
> |              | Llama-3-8B-instruct | Llama-3-70B-instruct | GPT-3.5-turbo-1106 |  GPT-4o  | Claude-3.5-sonnet | Average  |
> |:------------:|:-------------------:|:--------------------:|:------------------:|:--------:|:-----------------:|:--------:|
> |     CoA      |         25.5        |         18.8         |        15.5        |   17.5   |        3.4        |   16.14  |
> |  Crescendo   |         60.0        |         62.0         |        38.0        |   46.0   |        50.0       |   51.2   |
> | ActorAttack  |       **79.0**      |       **85.5**       |      **78.5**      | **84.5** |      **66.5**     | **78.8** |
>
> Reference:
> 1. Yang, Xikang, et al. "Chain of Attack: a Semantic-Driven Contextual Multi-Turn Attacker for LLM." arXiv preprint arXiv:2405.05610 (2024).

---

> ### Author Response · Authors · 2024-11-26
> **Author Rebuttal (Part two)**
>
> Q5: ***Comparison of state-of-the-art defense mechanisms***
>
> A5: Thank you for your valuable suggestion. In response, we have implemented and evaluated three distinct and powerful defense baselines: **Rephrase** [1], **RPO** [2], and **Circuit Breaker** [3], to comprehensively assess the effectiveness of our attack. Additionally, we included the performance of models fine-tuned using our multi-turn safety data. Since both Circuit Breaker and our defense mechanism rely on fine-tuning, we report the results specifically for Llama-3-8B-Instruct. The following table presents the Attack Success Rate (ASR) under these defenses.
>
>   Our findings show that Rephrase and RPO offer a partial reduction in ASR. However, Circuit Breaker greatly reduces the success rate of our attack, demonstrating the potential of safety alignment within the representation space. Moreover, fine-tuning with our safety data yields results comparable to **Circuit Breaker**,  highlighting the value and effectiveness of the safety data we have constructed.
>
>   We also note that since Circuit Breaker implements its defense using only single-turn safety data, we believe that integrating it with our multi-turn safety data could further improve model safety. We will include this result in our camera-ready version.
>
> |                                         | Llama-3-8b-instruct  | GPT-3.5-turbo-1106 | GPT-4o  |
> |:---------------------------------------:|:--------------------:|:------------------:|:-------:|
> |               No defense                |         78.0         |        78.5        |   84.5  |
> |                Rephrase                 |         54.0         |        50.0        |   48.0  |
> |                   RPO                   |         54.0         |        42.0        |   50.0  |
> |             Circuit breaker             |         28.0         |         -          |    -    |
> | Safety fine-tune using our safety data  |         32.0         |          -         |    -    |
>
> **Summary of Baselines**:
>   1. Rephrasing [1] is a system-level defense that paraphrases each query in our attacks. It is a simple yet surprisingly effective strategy against jailbreaks.
>   2. RPO [2] is a system-level defense method that optimizes safety suffixes. It has shown strong transferability across multiple LLMs and is effective against a variety of jailbreaks.
>   3. Circuit Breaker [3] is a state-of-the-art, model-level defense that interrupts the internal representations responsible for harmful outputs, effectively halting the generation of unsafe content.
>   We hope these results address your concerns and highlight the potential of our defense strategies in comparison to existing methods.
>
> References:
>   1. Neel Jain, Avi Schwarzschild, Yuxin Wen, Gowthami Somepalli, John Kirchenbauer, Ping yeh Chiang, Micah Goldblum, Aniruddha Saha, Jonas Geiping, and Tom Goldstein. Baseline defenses for adversarial attacks against aligned language models. arXiv:2309.00614, 2023.
>   2. Andy Zhou, Bo Li, and Haohan Wang. Robust prompt optimization for defending language models against jailbreaking attacks. arXiv preprint arXiv:2401.17263, 2024.
>   3. Andy Zou, Long Phan, Justin Wang, Derek Duenas, Maxwell Lin, Maksym Andriushchenko, Rowan Wang, Zico Kolter, Matt Fredrikson, and Dan Hendrycks. Improving alignment and robustness with circuit breakers. arXiv preprint arXiv:2406.04313, 2024.

---

> > ### Author Response · Authors · 2024-11-26
> > **Author Rebuttal (Part three)**
> >
> > Q6: ***Time cost and practicality of our method***
> >
> > A6: Thank you for this important question. We believe efficiency is a key strength of our method, and we would like to highlight several points regarding its practicality and time cost.
> >
> >   1. **About Practicality**: Apart from the victim model, the other three components of our attack—attacker, judge, and monitor—are implemented using the same LLM (GPT-4o) with our different instructions. Therefore, it is easy for users to deploy our attack via an API without additional setup or complex infrastructure.
> >
> >   2. **Efficiency Advantage**: One of the main advantages of our approach is its efficiency. First, our multi-turn attacks achieve high effectiveness without requiring feedback from the target model (as shown in Table 1). Our attacks show good transferability across many models without the need for additional adaptation. In contrast, existing methods like Crescendo rely on target model responses to dynamically modify their attacks paths. This reliance may introduce additional computational overhead when adapting to new models.
> >
> >   3. **Time Cost Comparison**: We also included comparisons with CoA and Crescendo in terms of time cost. However, due to differences in backend technologies (e.g., vLLM or Torch) and parallelization strategies used by these methods, a fair time cost comparison is challenging. Instead, we propose using the **average number of interactions with the target model per attack** as a more consistent efficiency metric. In this context, each turn of the attacks, including random trials, counts as one interaction. The results, shown in the table below, demonstrate that our method generally requires far fewer interactions to succeed compared to these baselines. Specifically, our approach achieves a **26% improvement in attack efficiency** over Crescendo, further confirming the efficiency advantages of our method.
> >
> > |             | Llama-3-8b-instruct | Llama-3-70b-instruct | GPT-3.5-turbo-1106 |  GPT-4o | Claude-3.5-sonnet | Average |
> > |:-----------:|:-------------------:|:--------------------:|:------------------:|:-------:|:-----------------:|:-------:|
> > |     CoA     |         14.6        |         13.6         |        15.8        |   14.6  |        43.3       |   20.4  |
> > |  Crescendo  |         10.5        |         10.3         |        12.0        |   11.5  |        14.9       |   11.8  |
> > | ActorAttack |       **8.3**       |        **8.0**       |       **8.5**      | **8.1** |      **10.9**     | **8.7** |
> >
> > Q7: ***Refinement of captions for figures 4(a) and 4(b)***
> >
> > A7: Thank you for your helpful suggestion! We have updated the captions for Figures 4(a) and 4(b) in the main text (Line 391) to make the image results clearer and more informative.

---

> > > ### Author Response · Authors · 2024-11-30
> > > **Follow-up on Rebuttal**
> > >
> > > Dear Reviewer,
> > >
> > > I hope you’re doing well. I have addressed the concerns raised in your initial review, and I would greatly appreciate your feedback on the revisions.
> > >
> > > Given the changes made, I kindly ask you to reconsider your score. If you need any further clarifications, please don’t hesitate to reach out.
> > >
> > > Thank you for your time and consideration.
> > >
> > > Best regards,
> > > Authors

---

### Official Review · Reviewer_8h4V · 2024-11-01

**Soundness:** 2
**Presentation:** 3
**Contribution:** 2
**Rating:** 5
**Confidence:** 4

**Summary:**

This paper proposes a multi-turn attack – ActorAttack, which is an improvement from the existing work. The main difference with previous multi-turn papers is the pre-set helpful clue, which is obtained by using LLM under diverse actors. With the designated clue, ActorAttack could generate the diverse attack paths towards the same harmful target.

**Strengths:**

1、The paper is well-arranged and readable
2、It provides reasonable experiment

**Weaknesses:**

1、	The innovation is limited
2、	Lack of in-depth root cause analysis
3、	The experiment result is not unconvincing enough

**Questions:**

1、	Limited novelty: The overall idea is basically consistent with crescendo: a) the generation of the initial clue, even if the actor theory is mentioned here, it actually only works in assigning a role to LLM for generating the initial clue; no strict theory or inspiration； b) In the initial generation process, the attack model in crescendo is simply replaced by the victim model; the benefits and reasons are unclear to confidently support the advantage; c) The dynamic multi-turn prompt modification action only involves repeated generation to reduce toxicity; no heuristic strategy guidance.
2、	Limited Contribution: As for multi-turn attack, this paper only contributes a point - designated clue
3、	Root cause analysis: Why multi-turn surpasses single-turn? Please give in-depth mechanism analysis and corresponding experiment support.
4、	Baseline Comparison: only one multi-turn baseline for comparison is not convincing, consider Yang, Xikang, et al. "Chain of Attack: a Semantic-Driven Contextual Multi-Turn attacker for LLM." arXiv preprint arXiv:2405.05610 (2024).
5、	Persuasive Experiment: a) only 50 instances and two LLMs (GPT-4o, Claude-3.5) in ablation study cannot support that your multi-turn attack – ActorAttack surpasses Crescendo, please add more results; b) only use GPT-4O to judge ASR is single-minded, consider human, Llama-guard model, and moderation API
6、	Explanation: Is it a common practice to use the embedding function of MiniLMv2 for diversity evaluation of the prompts? Give more evidence.
7、	kind Suggestion:
1)	Actor- involved in the multi-turn dialogue generation and update
2)	Add more strategical method to choose the most helpful clue
3)	The intuitive defense method could be improved

---

> ### Author Response · Authors · 2024-11-24
> **Author Rebuttal (Part One)**
>
> Thank you for your thoughtful and encouraging reviews! We are glad to hear that you found our paper well-arranged and readable, and appreciated the reasonableness of our experiments. We have carefully addressed your concerns below and look forward to continuing the discussion to clarify and strengthen our contributions.
>
> Q1: *Limited novelty: The overall idea is consistent with Crescendo.*
>
> A1: Thank you for your thoughtful observation. However, we respectfully disagree with the concern about the novelty and technical contribution of our work. **Grounded in sociological theory, our approach introduces a novel and systematic perspective to identify attack clues, enabling more diverse, effective, and efficient attacks compared to existing multi-turn baselines.**
>
> Key Contributions and Novelty
>
> We address two critical challenges for successful multi-turn attacks, which differentiate our work from prior approaches:
> 1. **Identifying Semantically Relevant and Diverse Attack Clues**
>   - Existing approaches either rely on limited strategies like role-playing and scenario assumptions (CoA [1] and CFA [2]) or fixed communication templates plus human-crafted seed instances (Crescendo [3]) to generate attacks. Limited attack strategies and potential biases towards seed instances may lead to the diversity issue, ultimately limiting their effectiveness (Table 1, and Figures 7 and 8).
>   - In contrast, our approach is systematically supported by scientific theory. Inspired by Latour’s sociological network theory, we explicitly model the potential attack clues as a network:
>     - **Nodes** represent attack clues that are semantically relevant to the harmful target.
>     - **Edges** form stepwise attack paths connecting clues to the harmful target.
>     - Our network has two distinct properties, which ensure the effectiveness of our attacks:
>       - **Higher diversity**: we identify six distinct types of nodes, including both human and non-human entities, covering a broader spectrum of attack clues.
>       - **Inherent semantic relevance**: different from existing works that infer attack clues from the seed instances, we generate a unique network for each harmful target, ensuring the derived clues are semantically relevant to the given target (Figure 2 and Figure 3(a)).
>     - Our quantitative results (Table 1 and Table 4) demonstrate the enhanced diversity and effectiveness of our approach.
> 2. **Efficient Convergence Toward the Harmful Target**
>   - Existing works like Crescendo reactively adjusts attack paths during the attack process based on the target model's responses. By contrast, we pre-compute the attack paths (i.e., each edge of our network serves the attack path) and the inherent semantic relevance of our attack clues ensures the convergence of our attack paths.
>   - Experimental results show that **our multi-turn attacks achieve high effectiveness without relying on feedback from the target model** (Table 1), validating the efficiency of our method. Another evidence is that under the same attack budget, our attacks consistently achieve greater effectiveness than Crescendo  (Figure 4). We also conducted a comparison between CoA and our method (see the table in response to Q5 from Reviewer 2ps2). These results demonstrate the efficiency of our method.
>
> In summary, our work introduces a novel perspective and addresses the limitations of previous methods, achieving state-of-the-art performance in multi-turn attacks. We hope this explanation clarifies our contributions and the distinction from Crescendo.

---

> ### Author Response · Authors · 2024-11-24
> **Author Rebuttal (Part Two)**
>
> Q2: ***Weak connection to theory.***
>
> A2: We respectfully disagree. Latour’s theory forms the foundation of our attack framework:
>
>  Based on Latour’s distinction between the roles of nodes in a network, we identified six categories of attack clues and different attack clues led to diverse attack paths (illustrated in Figure 11-14). Moreover, Latour emphasizes that both human and non-human actors hold equally significant positions in the network. This insight broadens our range of attack clues by including non-human entities such as books, media, or social movements.
>
> Therefore, this theoretical grounding is not merely a conceptual reference but a cornerstone of our methodology, enabling us to systematically and effectively model diverse attack clues.
>
> Q3: ***Lack of clarity about the reasons and benefits of initial attack generation.***
>
> A3: Thank you for pointing this out. The primary benefit of using the attacker model for initial attack generation is efficiency. Our attacks exhibit strong effectiveness even without relying on feedback from the target model (as evidenced in Table 1).
>
> Here’s why our method succeeds:
>
> - First, our network not only provides the attack clues (nodes), but also the attack paths (edges), which model the stepwise progression from a clue to the harmful target. Our pre-computed attack paths are model-agnostic.
> - Second, we hypothesize that LLMs trained on similar pre-training data (e.g., internet-crawled texts) exhibit consistent behavior, leading to similar responses to identical queries. Thus, we use the attacker model as the proxy of unknown target models and use a self-talk mechanism to generate initial attacks.
>
> We will clarify this explanation further in the camera-ready version.
>
> Q4: ***Dynamic modification only involves repeated generation to reduce toxicity, with no heuristic strategy guidance.***
>
> A4: Thank you for pointing this out. We would like to clarify that our dynamic modification strategy is not limited to toxicity reduction. Instead, we explicitly identified two classic cases in its design: (1) scenarios where the model refuses to answer due to toxicity concerns and (2) situations where the model is unaware of the answer to the given question. In the latter case, we discard the current attack path and re-sample a new one to restart the process (L249-256). **Our simple design leads to a non-trivial improvement in attack performance**, as demonstrated in Table 1. More importantly, the diversity and semantic relevance of our attack clues underpin the efficacy of our method, such that our dynamic modification does not need complex designs.
>
> Q5: ***Why multi-turn attacks are better than single-turn attacks.***
>
> A5: As discussed in our introduction, single-turn attacks often explicitly encode harmful intent in the input prompts (e.g., through adversarial suffixes [1], encodings [2], or role-playing [3]). Current safety alignment algorithms are adept at identifying and mitigating the harmful intent at a **single input level** [4, 5] but overlook the risks posed by **multi-turn contextual exploitation**.
> In contrast, our multi-turn attacks succeed by setting up a harmless yet semantically related conversational topic to conceal our harmful intention. Though our attack queries look innocuous, under our guidance, the model’s responses progressively reveal harmful details. Since our harmful intent is progressively expressed via contextual interactions, our attacks can bypass current safety mechanisms.
>
> References:
> 1. Andy Zou, Zifan Wang, Nicholas Carlini, Milad Nasr, J Zico Kolter, and Matt Fredrikson. Universal and transferable adversarial attacks on aligned language models. arXiv preprint arXiv:2307.15043, 2023.
> 2. Alexander Wei, Nika Haghtalab, and Jacob Steinhardt. Jailbroken: How does llm safety training fail? Advances in Neural Information Processing Systems, 36, 2024.
> 3. Patrick Chao, Alexander Robey, Edgar Dobriban, Hamed Hassani, George J. Pappas, and Eric Wong. Jailbreaking black box large language models in twenty queries. arXiv preprint arXiv:2310.08419, 2024.
> 4. Long Ouyang, Jeffrey Wu, Xu Jiang, Diogo Almeida, Carroll Wainwright, Pamela Mishkin, Chong Zhang, Sandhini Agarwal, Katarina Slama, Alex Ray, et al. Training language models to follow instructions with human feedback. Advances in neural information processing systems, 35: 27730–27744, 2022.
> 5. Yuntao Bai, Andy Jones, Kamal Ndousse, Amanda Askell, Anna Chen, Nova DasSarma, Dawn Drain, Stanislav Fort, Deep Ganguli, Tom Henighan, et al. Training a helpful and harmless assistant with reinforcement learning from human feedback. arXiv preprint arXiv:2204.05862, 2022.

---

> ### Author Response · Authors · 2024-11-24
> **Author Rebuttal (Part three)**
>
> Q6: ***Including more multi-turn baselines for comparison.***
>
> A6: Thank you for the suggestion! We have included CoA as a baseline to further validate the effectiveness of our method. The results, as shown in the table, demonstrate that our ActorAttack significantly outperforms CoA against both open-source and closed-source LLMs on Harmbench.
>
> |              | Llama-3-8b-Instruct | GLM4-9b-instruct | Qwen2.5-7b-instruct | GPT-4o   | Claude-3.5-sonnet |
> |:--------------:|:---------------------:|:------------------:|:---------------------:|:----------:|:-------------------:|
> | CoA          | 25.5                | 26.0             | 25.0                | 17.5     | 3.4               |
> | ActorAttack  | **79.0**            | **73.0**         | **83.5**            | **84.5** | **66.5**          |
>
> Q7: ***Non-adequate evaluations in ablation study.***
>
> A7: Thank you for raising this concern. We selected GPT-4o and Claude-3.5 as target models because they are currently among the safest LLMs based on evaluations over established safety benchmarks such as Harmbench and Sorrybench. Additionally, due to the high API costs associated with using Crescendo, we conducted evaluations on a uniformly sampled subset of 50 instances from Harmbench rather than the full set.
>
> To address the reviewer's concern, we expanded our evaluation by including two additional LLMs Llama-3-8B-instruct and Qwen-2.5-7b-instruct, and assessing the complete Harmbench dataset. The results, presented in the table, confirm our findings: ActorAttack consistently demonstrates higher effectiveness and better diversity compared to Crescendo.
>
> | Number of queries | 2        | 3        | 4        | 5        |
> |-------------------|----------|----------|----------|----------|
> | Crescendo         | 20.0     | 44.0     | 48.0     | 60.0     |
> | ActorAttack       | **74.0** | **79.0** | **84.5** | **85.0** |
> - table 1. results of Llama-3-8b-instruct
>
> | Number of queries | 2        | 3        | 4        | 5        |
> |-------------------|----------|----------|----------|----------|
> | Crescendo         | 24.0     | 40.0     | 54.0     | 69.0     |
> | ActorAttack       | **65.0** | **79.0** | **83.5** | **87.5** |
> - table 2. results of Qwen-2.5-7b-instruct
>
> Q8: ***The rationality of using GPT-4o for judgment.***
>
> A8: Thank you for this suggestion! We believe that our choice of GPT-4o as the judgment model is reasonable and well-supported. Our design aligns with the practices of [1], [2], and [3], which implement GPT-4-based judges. Notably, [1] found through human studies that GPT-4 is effective and accurate in identifying harmful outputs.
> To further validate the rationality of using GPT-4o as the judge, we conducted additional human study experiments. As shown in the table below, our findings confirm that the GPT-4o judge aligns more closely with human judgments compared to alternatives like Llama-Guard and the OpenAI Moderation API. Specifically, Llama-Guard exhibits a higher false negative rate (misclassifying unsafe outputs as safe), while the OpenAI Moderation API shows a higher false positive rate (misclassifying safe outputs as unsafe). These results underscore the reliability and alignment of GPT-4o for this task.
>
> |                       | Llama-3-8b-instruct | Qwen2.5-7b-instruct | GLM4-9b-instruct | GPT-4o        | Claude-3.5-sonnet |
> |-----------------------|---------------------|---------------------|------------------|---------------|-------------------|
> | Human                 | 80%                 | 82%                 | 82%              | 88%           | 70%               |
> | GPT-4o                | **74% (-6%)**       | **78% (-4%)**       | **74% (-8%)**    | **82% (-6%)** | **74% (+4%)**     |
> | Llama-guard           | 58% (-32%)          | 48% (-34%%)         | 94% (+12%)       | 60% (-28%)    | 56% (-14%)       |
> | OpenAI Moderation API | 100%( +20%)         | 96% (+14%)          | 100% (+18%)      | 94% (+6%)     | 96% (+26%)        |
>
> References:
> 1. Xiangyu Qi, Yi Zeng, Tinghao Xie, Pin-Yu Chen, Ruoxi Jia, Prateek Mittal, and Peter Henderson. Fine-tuning aligned language models compromises safety, even when users do not intend to! arXiv preprint arXiv:2310.03693, 2023.
> 2. Yi Zeng, Hongpeng Lin, Jingwen Zhang, Diyi Yang, Ruoxi Jia, and Weiyan Shi. How johnny can persuade llms to jailbreak them: Rethinking persuasion to challenge ai safety by humanizing llms. arXiv preprint arXiv:2401.06373, 2024.
> 3. Qibing Ren, Chang Gao, Jing Shao, Junchi Yan, Xin Tan, Wai Lam, and Lizhuang Ma. Exploring safety generalization challenges of large language models via code. In The 62nd Annual Meeting of the Association for Computational Linguistics, 2024.

---

> > ### Author Response · Authors · 2024-11-24
> > **Author Rebuttal (Part four)**
> >
> > Q9. ***The soundness of diversity evaluation***
> > A9: Using BERT-sentence embedding models for diversity assessment is an established practice recommended in [1] [2] [3]. As for the specific model choice, we follow [4] to select MiniLMv2, a powerful sentence embedding model to capture the semantic information of sentences [5].
> >
> > Q10: ***Kind suggestions from the reviewer***
> >
> > Q10-1: ***Actor update in one multi-turn attack***
> >
> > A10-1: Thank you for this suggestion! We’re not sure we fully understand your idea here. Currently, we select one clue to generate one multi-turn attack sequence. Do you mean using different actors within one multi-turn attack?
> >
> > Q10-2 ***Exploring the way of selecting the attack clue***
> >
> > A10-2: This is an exciting direction! In our current approach, we propose filtering out attack clues when the target model does not know the answer to queries derived from these clues. Additionally, we see potential in exploring a more sophisticated selection strategy, such as greedily choosing specific types of attack clues based on historical data to improve attack efficiency. Unfortunately, we cannot incorporate these additional experiments in the current version due to time constraints. However, we plan to investigate this thoroughly and report results in the camera-ready paper.
> >
> > Q10-3. ***Improve the defense method***
> >
> > A10-3: We appreciate your valuable suggestion! Currently, the main challenge is the lack of open-source multi-turn safety alignment datasets. Therefore, we construct the first multi-turn safety dataset to fill this gap, and we demonstrate LLMs trained using our dataset are more robust to multi-turn attacks. We plan to leave more advanced algorithm designs for future work.
> >
> > References:
> >
> > 1. Ethan Perez, Saffron Huang, Francis Song, Trevor Cai, Roman Ring, John Aslanides, Amelia Glaese, Nat McAleese, and Geoffrey Irving. Red Teaming Language Models with Language Models.  arXiv preprint arXiv:2202.03286, 2022.
> > 2. Guy Tevet and Jonathan Berant, Evaluating the Evaluation of Diversity in Natural Language Generation, arXiv preprint arXiv:2004.02990, 2021.
> > 3. Zhang-Wei Hong, Idan Shenfeld, Tsun-Hsuan Wang, Yung-Sung Chuang, Aldo Pareja, James Glass, Akash Srivastava, and Pulkit Agrawal. Curiosity-driven red-teaming for large language models. arXiv preprint arXiv:2402.19464, 2024.
> > 4. Seanie Lee, Minsu Kim, Lynn Cherif, David Dobre, Juho Lee, Sung Ju Hwang, Kenji Kawaguchi, Gauthier Gidel, Yoshua Bengio, Nikolay Malkin, et al. Learning diverse attacks on large language models for robust red-teaming and safety tuning. arXiv preprint arXiv:2405.18540, 2024.
> > 5. Wenhui Wang, Hangbo Bao, Shaohan Huang, Li Dong, and Furu Wei. Minilmv2: Multi-head self-attention relation distillation for compressing pretrained transformers. arXiv preprint arXiv:2012.15828, 2020.

---

> > > ### Author Response · Authors · 2024-11-30
> > > **Follow-up on Rebuttal**
> > >
> > > Dear Reviewer,
> > >
> > > I hope you’re doing well. I have addressed the concerns raised in your initial review, and I would greatly appreciate your feedback on the revisions.
> > >
> > > Given the changes made, I kindly ask you to reconsider your score. If you need any further clarifications, please don’t hesitate to reach out.
> > >
> > > Thank you for your time and consideration.
> > >
> > > Best regards,
> > > Authors

---

### Note · Authors · 2024-12-16

I have read and agree with the venue's withdrawal policy on behalf of myself and my co-authors.